# The Relationship between Clock Genes, Sirtuin 1, and Mitochondrial Activity in Head and Neck Squamous Cell Cancer: Effects of Melatonin Treatment

**DOI:** 10.3390/ijms241915030

**Published:** 2023-10-09

**Authors:** César Rodríguez-Santana, Alba López-Rodríguez, Laura Martinez-Ruiz, Javier Florido, Olga Cela, Nazzareno Capitanio, Yolanda Ramírez-Casas, Darío Acuña-Castroviejo, Germaine Escames

**Affiliations:** 1Biomedical Research Center, Health Sciences Technology Park, University of Granada, 18016 Granada, Spain; cesar@correo.ugr.es (C.R.-S.); albalopezrodriguez4@gmail.com (A.L.-R.); lauramartinezr@ugr.es (L.M.-R.); javiflorido@ugr.es (J.F.); yolandaramirez@correo.ugr.es (Y.R.-C.); dacuna@ugr.es (D.A.-C.); 2Department of Physiology, Faculty of Medicine, University of Granada, 18071 Granada, Spain; 3Department of Clinical and Experimental Medicine, University of Foggia, 71122 Foggia, Italy; olga.cela@unifg.it (O.C.); nazzareno.capitanio@unifg.it (N.C.); 4Centro de Investigación Biomédica en Red Fragilidad y Envejecimiento Saludable (CIBERFES), Instituto de Investigación Biosanitaria (Ibs), San Cecilio University Hospital, 18016 Granada, Spain

**Keywords:** head and neck squamous cell carcinoma, chronodisruption, clock genes, melatonin, mitochondria

## Abstract

The circadian clock is a regulatory system, with a periodicity of approximately 24 h, which generates rhythmic changes in many physiological processes, including mitochondrial activity. Increasing evidence links chronodisruption with aberrant functionality in clock gene expression, resulting in multiple diseases such as cancer. Melatonin, whose production and secretion oscillates according to the light–dark cycle, is the principal regulator of clock gene expression. In addition, the oncostatic effects of melatonin correlate with an increase in mitochondrial activity. However, the direct links between circadian clock gene expression, mitochondrial activity, and the antiproliferative effects of melatonin in cancers, including head and neck squamous cell carcinoma (HNSCC), remain largely unknown. In this study, we analyzed the effects of melatonin on HNSCC cell lines (Cal-27 and SCC9), which were treated with 500 and 1000 µM melatonin. We found that the antiproliferative effect of melatonin is not mediated by the *Bmal1* clock gene. Additionally, high doses of melatonin were observed to result in resynchronization of oscillatory circadian rhythm genes (*Per2* and *Sirt1*). Surprisingly, the resynchronizing effect of melatonin on *Per2* and *Sirt1* did not produce alterations in the oscillation of mitochondrial respiratory activity. These results increase our understanding of the possible antiproliferative mechanisms in melatonin in the treatment of head and neck squamous cell carcinoma and suggest that its antiproliferative effects are independent of clock genes but are directly related to mitochondrial activity.

## 1. Introduction

Head and neck squamous cell carcinoma (HNSCC) is a prevalent and aggressive malignancy that arises from the epithelial cells lining the upper aerodigestive tract, including the oral cavity, pharynx, larynx, and nasal cavity. HNSCC is notorious for its aggressive behavior and high recurrence rates even after successful treatment. For this reason, an understanding of the underlying mechanisms driving HNSCC development and progression is crucial for improving therapeutic approaches and patient outcomes [1,2].

Emerging evidence suggests that disturbances in circadian rhythms and the circadian clock, known as chronodisruption, may play a critical role in HNSCC development, progression, and therapeutic outcomes [3,4,5]. At the molecular level, the circadian clock is composed of interlocked cycles of auto-regulatory transcription and translation feedback loops. These loops are operated by the transcriptional activators BMAL1 and CLOCK, which promote various clock-controlled genes such as Period 1, 2, 3 (*Per 1–3*) and Cryptochrome 1, 2 (*Cry 1, 2*) in order to induce their transcription. Subsequently, proteins PER and CRY inhibit CLOCK/BMAL1-mediated transcription in a negative feedback manner. However, the orphan nuclear receptors REV-ERBα and RORα operate a feedback loop that negatively and positively controls *Bmal1* transcription [6]. In addition, in many clock genes, the amplitude of oscillation is influenced by SIRT1, a type III histone/protein deacetylase. Moreover, SIRT1 is involved in regulating mitochondrial bioenergetics and cell metabolism. Recent studies have consistently reported the existence of a nicotinamide phosphoribosyltransferase (NAMPT)–nicotinamide adenine dinucleotide (NAD)–SIRT1 axis, whose expression exhibits circadian rhythmicity. Therefore, the role played by mitochondrial bioenergetics and dynamics in cell metabolism highlights the interplay between mitochondrial oxidative phosphorylation (OxPhos) and clock genes [7,8].

Melatonin (N-acetyl-5-methoxytryptamine, aMT) is the principal hormone that regulates circadian clock genes [9]. Although melatonin is produced by the pineal gland, higher concentrations are also produced in other tissues [10]. As the synthesis of pineal melatonin is associated with light/dark photoperiods, its production follows a circadian rhythm and constitutes a chronobiotic signal that synchronizes rhythms. Moreover, the alteration of melatonin production is associated with an increased incidence of cancer, which could correlate with the decreased regulation of the circadian machinery [11,12,13].

Recently, several studies have reported that high concentrations of melatonin have numerous oncostatic effects with no side effects [14,15,16,17]. Melatonin targets mitochondria and improves their functioning by enhancing mitochondrial OxPhos and by increasing ATP production in nontumor cells. However, this does not occur in tumor cells, thus leading to decreased cell proliferation [18,19]. In this context, we have previously demonstrated that melatonin reverses metabolic reprogramming and decreases proliferation in HNSCC cell lines Cal-27 and SCC9 [15,20]. This hormone is therefore of particular importance for the development of innovative cancer treatments. In this study, we hypothesized that melatonin is capable of countering cell proliferation in HNSCC by resynchronizing the deregulated circadian machinery of these cancer cells by focusing on the interplay between mitochondrial function and the functioning of the biological clock.

## 2. Results

### 2.1. High Concentrations of Melatonin Significantly Affected the Expression of Clock Genes Bmal1 and Per2 in HNSCC Cells

Studies have shown that certain circadian genes are altered in some cancers [21,22,23]. Moreover, previous studies have demonstrated that only high doses of melatonin lead to HNSCC apoptosis due to excessive ROS production [15]. Previously, we performed a dose–response study, and our data suggested that there is a correlation between the cell content of melatonin and its apoptotic effects, thus supporting the notion that high concentrations of melatonin in cancer cells are required for its cytotoxic effects [15]. However, the expression patterns of clock genes have not been studied at high doses of melatonin. Therefore, we first analyzed the expression of the clock genes *Bmal1* and *Per2* in Cal-27 and SCC9 cells treated with vehicle (control) or melatonin (500 and 1000 µM) at different time points during a period of 24 h. In addition, we established another experimental group, in which Cal-27 and SCC9 cells in culture were exposed to the well-established serum shock protocol to reset the clockwork machinery [24], thus facilitating the homogeneity of clock gene expression once the repressive condition is removed (Figure 1). The statistical data relating to the cosinor analysis of clock gene rhythms, including their significance, acrophase, and amplitude, are shown in Table 1.

Regarding *Bmal1*, both cell lines showed a circadian basal expression in the control group. However, following synchronization with serum shock, the expression of *Bmal1* showed significant differences with respect to the control. The Cal-27 cells displayed a broader amplitude, while SCC9 also showed an earlier acrophase. The cells treated with melatonin did not show differences with respect to the control except at a dosage of 1000 µM, at which melatonin induced a loss of circadian rhythm in *Bmal1* (Table 2).

The rhythm of the other regulator gene *Per2* was also detected after serum shock in both cell lines, Cal-27 and SCC9. Nevertheless, no circadian rhythm was detected in the control group, without serum shock. However, 1000 µM melatonin induced circadian rhythmicity with respect to *Per2* in both cell lines. In addition, the comparison between serum shock and melatonin 1000 µM showed a significant difference in the acrophase (Table 2). The expression of *Per2* has an inhibitory effect on *Bmal1*. This would explain the effect observed at aMT 1000 µM on Cal-27 (Figure 1A) upon induction of a circadian rhythm by Per2 at aMT 1000 µM (Figure 1B). However, this does not occur in SCC9, further suggesting a clock gene-independent function of *Per2*.

### 2.2. The expression of Bmal1 Is Not Affected by Low Doses of Melatonin in HNSCC Cells

Since melatonin causes resynchronization of the various clock genes in non-cancer cells at low doses, we analyzed the effects of melatonin at 10 nM and 100 µM on Cal-27 cells (Appendix A). The results did not show significant differences with respect to the control group. Therefore, as shown in Appendix A, we concluded that low doses of melatonin did not have any significant effect on the circadian rhythmicity of *Bmal1* in HNSCC cells (Figure 2).

### 2.3. High Doses of Melatonin Induce the Circadian Expression of Sirtuin-1 in HNSCC Cells

Sirt1 plays a key role in the regulation of *Per2* circadian expression. In addition, in recent years, the NAMPT–NAD–SIRT1 axis and mitochondrial metabolism have aroused scientific interest. For these reasons, we studied the effect of melatonin on *Sirt1* expression (Table 3). The data demonstrate that 500 µM and 1000 µM doses of melatonin induced synchronization of the *Sirt1* circadian rhythm as compared to the control group (Figure 3). In addition, melatonin induced a significantly advanced acrophase as compared to that obtained after serum shock (Table 4). This confirms that melatonin at high doses directly affects *Sirt1* expression in HNSCC cells.

### 2.4. High Doses of Melatonin Have No Effect on the Induction of the Circadian Rhythm of Mitochondrial Respiratory Activity in Cal-27 Cells

Next, we decided to determine whether the changes in the expression of *Bmal1* and *Per2,* as well as *Sirt1,* induced by melatonin, give rise to an alteration in mitochondrial respiratory activity. On the other hand, we also tried to determine whether an alteration in mitochondrial respiratory activity is responsible for an alteration in the expression of the genes mentioned. Thus, the Cal-27 cells were subjected to the serum shock protocol in order to facilitate the homogenous synchronization of clock gene expression. Table 5 shows the results of a systematic analysis, whereby cellular respiration in intact cells was assessed every three hours following synchronization. The OCR was measured, as described in Section 4.5. Figure 4 shows that the Cal-27 cells, following synchronization, displayed a circadian rhythm of respiratory activity which peaked at h 17. However, we did not observe any changes in the rhythmicity of mitochondrial respiratory activity in either the control group or in the groups treated with high concentrations of melatonin (Figure 4).

To rule out the possibility that mitochondrial respiratory activity was affected after high doses of melatonin treatment, we synchronized the cells following serum shock, as well as in the control and after 1000 μM melatonin treatment. Figure 5 shows similar results for the group following serum shock. Respiratory activity showed a circadian rhythmicity (Appendix A). These data demonstrate that high doses of melatonin do not alter mitochondrial respiratory capacity (Table 6).

### 2.5. Bmal1 Knockdown Increases Cell Proliferation While Melatonin’s Antiproliferative Effects Are Maintained

Given that Bmal1, a major link in mitochondrial activity, plays a key role in cell proliferation, we decided to study the antiproliferative effect of melatonin by reducing the expression of this clock gene. Figure 6 shows that, following the reduction in *Bmal1* expression by RNA interference, the proliferation of Cal-27 and SCC9 is significantly increased. However, the antiproliferative effects of melatonin continued to be observed in a dose–response manner in both control and knockdown cells.

## 3. Discussion

Previous studies have demonstrated that high concentrations of melatonin are required to exert its antiproliferative effects in HNSCC [15,25]. However, the effects of high doses of melatonin on clock genes in these cancer cells have not been studied. Very few reports have highlighted the possible effects of melatonin on the circadian rhythm clock components in HNSCC cells [26]. The present study shows, for the first time, the results of high doses of melatonin on clock gene expression in HNSCC cells. We have demonstrated that the antiproliferative effect of melatonin is not mediated by the clock gene *Bmal1* and that melatonin treatment results in resynchronization of the oscillatory circadian rhythm of the *Per2* and *Sirt1* genes. However, these changes in clock genes and in *Sirt1,* produced by melatonin, did not affect oscillations in mitochondrial respiratory activity. Having recently shown that the antiproliferative action of melatonin on HNSCC cells depends on its interaction with mitochondria [15], our results suggest that clock genes are not involved in the cytotoxic activity of melatonin in HNSCC cell lines Cal-27 and SCC9. Therefore, this study has made it possible to explore how high doses of melatonin affect the cellular circadian machinery independently of its relationship with mitochondrial metabolism.

Currently, there is considerable scientific evidence that shows alterations in the circadian machinery in a variety of tumors, including head and neck cancer [5,27,28]. Moreover, alterations in *Bmal1* play a critical role in the regulation of cell proliferation in cancers, which are linked to an acceleration in tumor development and modifications in responses to anti-cancer drugs [5,29]. Moreover, the expression profiles of circadian rhythm components in cancers, as well as their alteration, depend on the patient involved and/or the cell line. In this study, we demonstrated the presence of circadian expression of *Bmal1* in Cal-27 and SCC9 cells. We also show that melatonin, at high concentrations, did not affect the resynchronization of *Bmal1*, although the rhythm of *Bmal1* was blunted at the melatonin dose of 1000 µM in Cal-27 with no effect on SCC9, possibly due to these cells being more sensitive to melatonin (Appendix A) [15,20]. Surprisingly, at levels closer to physiological doses (10 nM and 100 µM), melatonin does not resynchronize *Bmal1*, thus suggesting that low levels of melatonin have no effect on cancer clock genes in HNSCC cells. In addition, we recently demonstrated that high doses of melatonin significantly increased MT1 gene expression and also led to a marked reduction in MT2 and RORα expression [20]. These results reinforce the notion that high doses of melatonin do not disrupt receptor expression but may play a role, at least partly, in its impact on cancer cells. Our results have therefore challenged the hypothesis that melatonin has an antiproliferative effect only at high doses.

To gain an insight into the antiproliferative effects of melatonin and given the importance of *Bmal1* in the regulation of the cell cycle and apoptosis [30,31], we decided to inhibit this clock gene via RNA interference. Surprisingly, in both the knockdown cell lines used in our experiments, the antiproliferative impact of melatonin was quite similar in both the presence and absence of *Bmal1*. This finding demonstrates that the antiproliferative effect of melatonin in HNSCC cells is independent of *Bmal1* in HNSCC cells.

Furthermore, alterations in Per2 have also been related to different types of cancers such as lung cancer [32], breast cancer [33], and HNSCC [5]. In this study, we demonstrate that *Per2* expression in HNSCC does not exhibit a circadian rhythm (Figure 1). However, the synchronization of Cal-27 and SCC9 cells induced circadian expression in *Per2*. These findings enabled us to determine that circadian machinery is present in the two cell lines studied, which lack regulated *Per2* clock gene expression, however. Interestingly, treatment with melatonin (1000 µM) altered *Per2* levels by resynchronizing the rhythmic gene expression pattern in HNSCC. These findings are consistent with Jung et al. (2010) [26], who showed that high concentrations of melatonin induce rhythmic *Per2* expression in prostate cancer cells. Per2 is an important tumor suppressor that regulates apoptosis by upregulating p53 and BAX and downregulating c-Myc and Bcl-2 [34]. These findings could facilitate the provision of additional data to bolster the pro-apoptotic effect of melatonin in cancer cells.

An additional unexpected finding of this study is that mitochondrial respiratory activity does not fluctuate in a circadian manner following treatment with high concentrations of melatonin. This is of particular importance given the resynchronization of *Sirt1* expression observed after treatment with melatonin (Figure 3). Numerous studies have addressed the role played by mitochondrial bioenergetics and dynamics in cell metabolism in relation to circadian rhythmicity, which is mainly due to PGC-1α, the target of Sirt1 [35,36,37]. In addition, Cela et. al. (2016) [8] have shown that rhythmic respiratory activity is associated with the NAMPT–NAD–Sirt1 axis and the oscillating acetylation/deacetylation state of complex I. However, we found that the circadian oscillations observed in *Sirt1* caused by melatonin do not translate into oscillations in mitochondrial respiratory activity, as was observed in serum shock-treated cells. In this regard, the serum shock protocol enables the clockwork machinery to be reset, which facilitates the homogenous synchronization of cellular clock gene expression in culture once the repressing condition is released [24]. We can therefore conclude that melatonin exerts a specific effect on the expression of *Sirt1* which does not translate into mitochondrial changes.

## 4. Materials and Methods

### 4.1. Cell Culture and Treatment

The head and neck squamous carcinoma cells, Cal-27 and SCC9, were obtained from the Cell Bank at the Scientific Instrumentation Centre of the University of Granada (ATCC: CRL2095 and CRL1629, respectively). The Cal-27 cells were maintained in Dulbecco’s modified Eagle’s medium high glucose (DEMEM; DMEM-HHSTA; Capricorn Scientific GmbH, Ebsdorfergrund, Germany) supplemented with 10% fetal bovine serum (FBS; 16000044; ThermoFisher Scientific, Waltham, MA, USA), 2% antibiotic/antimycotic solution (15240062; ThermoFisher Scientific), and 1 mM sodium pyruvate (S8636-100ML; Sigma, Madrid, Spain). The SCC9 cells were grown in DMEM-F12 (11320033; 1:1; ThermoFisher Scientific) Nutrient Mixture Ham’s medium containing 10% FBS, 2 mM L-glutamine (25030081; ThermoFisher Scientific), 0.4 μg/mL hydrocortisone (H0888; Sigma), 2% antibiotic/antimycotic solution, and 0.5 mM sodium pyruvate. Both cell lines were maintained at 37 °C in a humidified atmosphere of 5% CO_2_ and 95% air.

Melatonin (Fagron Ibérica S.A.U., Zaragoza, Spain) stock solution was prepared in 15% propylene glycol (PG; 24414.296; VWR, Radnor, PA, USA) in phosphate-buffered solution (PBS; 14190-094; Life Technologies, Carlsbad, CA, USA) and filter-sterilized through a 0.2 μm-pore filter (Sartorius Biotech GmbH, Gottingen, Germany). The cells were grown to 60–70% confluence and the serum was starved for 48 h. They were then treated with different concentrations of melatonin (500 and 1000 µM). The cells were harvested and assayed at the different time points indicated in the figures and the text. Vehicle was added to the control group.

### 4.2. Cell Proliferation Assay

Cell counts at the end of treatment were carried out using the CyQuant cell proliferation assay (C7026, ThermoFisher Scientific, Waltham, MA, USA). This assay, which has a linear detection range of 50–50,000 cells/well, is dependent on a green dye (CyQuant-GR) that fluoresces when bound to cellular nucleic acids. At the end of the treatment, a CyQuant assay was performed according to the manufacturer’s instructions. Fluorescence was measured (excitation, 480 nm; emission, 520 nm) using an FLx800 microplate fluorescence reader (Bio-Tek Instruments Inc., Winooski, VT, USA) and was then compared with a standard curve to determine the number of cells.

### 4.3. Quantitative RT-PCR

Total RNA from Cal-27 and SCC-9, at different time points, was extracted using the RNeasy^®^ Plus Mini Kit (74136, Qiagen, Hilden, Germany) according to the manufacturer’s instructions. cDNA was synthesized from 400 ng total RNA (95047, Quantabio, Beverly, MA, USA). For real-time PCR, we used PerfeCTa^®^ SYBR^®^ Green FastMix™ (95074-012, Quantabio, Beverly, MA, USA). The primer sequences are shown in Table 7. The target gene expression levels were normalized using the housekeeping control gene *GAPDH*. The amount of mRNA for each target gene relative to *GAPDH* was calculated using the comparative Ct (or 2^(−ΔΔCt)^) method.

### 4.4. Respirometric Measurements

The cultured cells were gently detached from the dish using the tripsinisation process, washed in PBS, harvested by centrifugation at 800× *g* for 5 min, and then immediately assessed for O_2_ consumption with the aid of an Oxygraph-2 k high-resolution oximeter (Oroboros Instruments, Innsbruck, Austria). Approximately 1 × 10^7^ viable cells/mL were assayed in DMEM-F12 medium at 37 °C. After achieving a stationary endogenous substrate-sustained resting oxygen consumption rate (OCR), this rate was corrected for 2 μM antimycin A plus 2 μM rotenone-insensitive respiration and then normalized to the initial cell number, with viability determined using trypan blue staining.

### 4.5. BMAL1-Specific siRNA Transfection in Cal-27 Cells

BMAL1-specific siRNA was purchased from Sigma-Aldrich (St. Louis, MO, USA, Mission Pre-designed siRNA2D). The Cal-27 cells were seeded on 24-well plates at 60–70% confluence and were transiently transfected with the BMAL1-specific siRNA diluted in Opti-MEM using Lipofectamine™ RNAiMAX Transfection Reagent (13778075, ThermoFisher Scientific, Waltham, MA, USA) according to the manufacturer’s protocol. After 24 h of incubation at 37 °C, the transfection medium was replaced with complete medium containing 10% FBS, and the experiments were conducted on the basis of the treatments carried out in the previous experiments.

### 4.6. Statistical Analysis

Statistical analyses were carried out using GraphPad Prism v. 8.0.1 software (GraphPad Software Inc., La Jolla, CA, USA). The data were expressed as the mean standard error of the mean (S.E.M) of at least three independent experiments. Unpaired Student’s *t*-tests were used to compare differences between the experimental groups and their respective untreated controls. A *p*-value < 0.05 was statistically significant. A rhythmicity analysis of the experimental data was carried out using the non-linear cosinor regression tool CircaCompare when the parameters significantly oscillated in the two groups [38]. We assumed a period of 24 h, while the rhythm detection was considered statistically significant at a *p*-value < 0.05 with confidence levels of 95%.

## 5. Conclusions

In this study, we examined whether the antiproliferative activity of melatonin in HNSCC depends on clock genes. To this end, we explored the circadian machinery and its connection with mitochondrial respiratory activity. We were able to confirm that the antiproliferative effect of melatonin is independent of *Bmal1* and that melatonin is capable of resynchronizing *Per2* and *Sirt1* rhythmicity. However, these effects of melatonin on the circadian machinery do not alter the rhythmicity of mitochondrial respiratory activity. Overall, our findings suggest that the oncostatic effects of melatonin are exerted independently of the circadian clock genes and confirm previous data pointing to the involvement of a direct effect on the mitochondria. This study constitutes a novel approach to gaining a better understanding of melatonin’s oncostatic mechanisms.

## Figures and Tables

**Figure 1 ijms-24-15030-f001:**
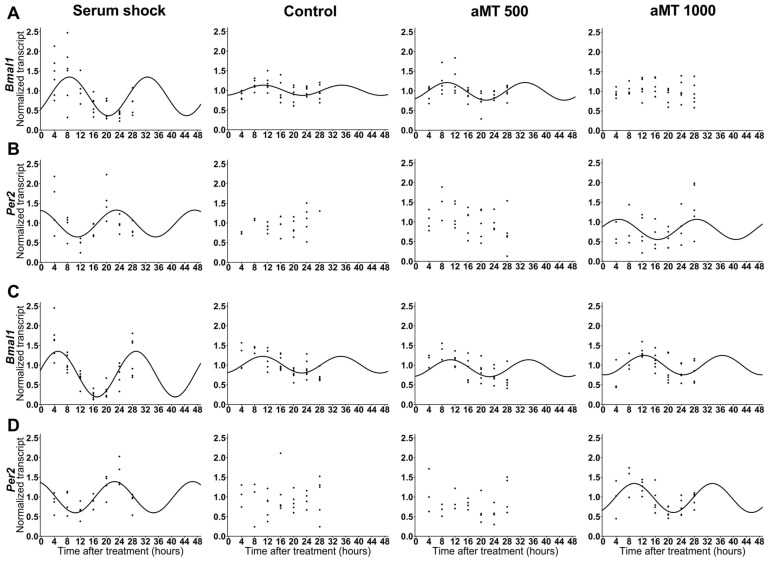
Relative expression of clock genes in HNSCC cell lines Cal-27 (**A**,**B**) and SCC9 (**C**,**D**): (**A**,**C**) *Bmal1* and (**B**,**D**) *Per2* after serum shock, control, and aMT (500 and 1000 µM) treatments; *n* = 3–6 independent experiments. The best cosinor fit is shown as a continuous line at a time of 48 h.

**Figure 2 ijms-24-15030-f002:**
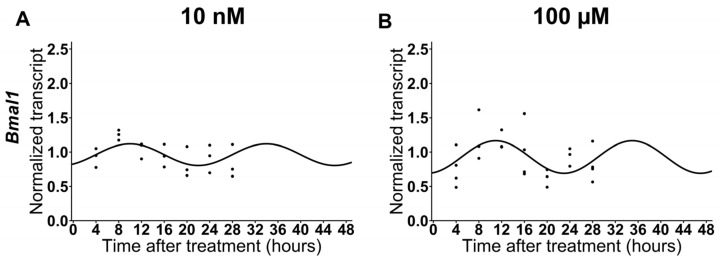
Relative expression of *Bmal1* in Cal-27 cells with 10 nM aMT (**A**) and 100 µM aMT (**B**) treatments; *n* = 3–4 independent experiments. The best cosinor fit is shown as a continuous line at a time of 48 h.

**Figure 3 ijms-24-15030-f003:**
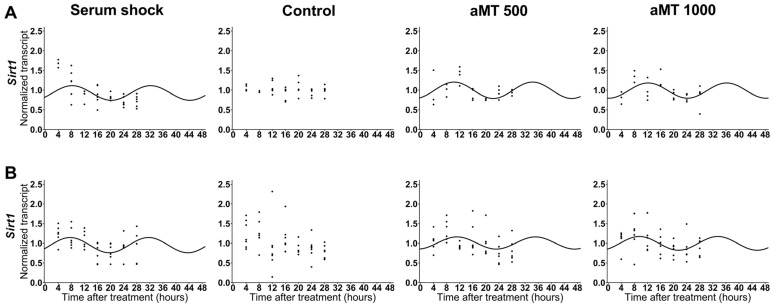
Relative expression of *Sirt1* in HNSCC cell lines Cal-27 (**A**) and SCC9 (**B**) after serum shock, control, and aMT (500 and 1000 µM) treatments; *n* = 3–6 independent experiments. The best cosinor fit is shown as a continuous line at a time of 48 h.

**Figure 4 ijms-24-15030-f004:**
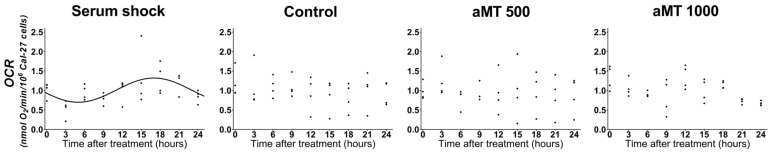
Measurement of mitochondrial respiratory activity in intact Cal-27 cells after serum shock, control, and aMT (500 and 1000 µM) treatments. OCR refers to the resting condition (i.e., endogenous substrate-sustained respiration); *n* = 3–6 independent experiments. The best cosinor fit is shown as a continuous line.

**Figure 5 ijms-24-15030-f005:**
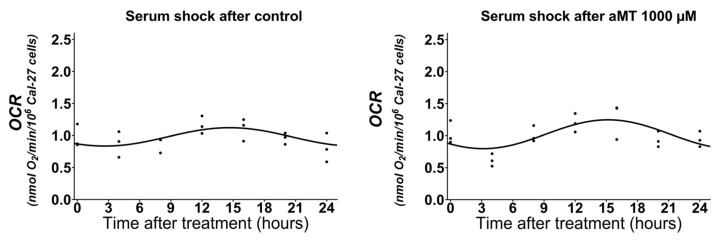
Measurement of mitochondrial respiratory activity in intact Cal-27 cells after serum shock, in control and aMT 1000 µM. OCR refers to resting the condition (i.e., endogenous substrate-sustained respiration); *n* = 3 independent experiments. The best cosinor fit is shown as a continuous line.

**Figure 6 ijms-24-15030-f006:**
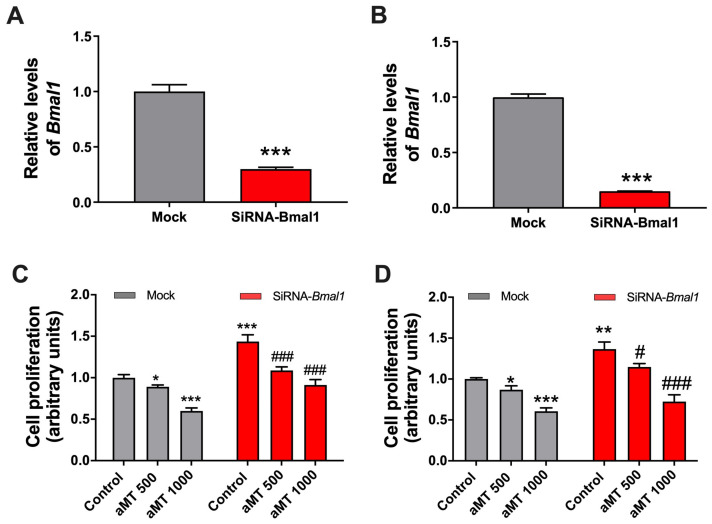
Effect of melatonin on *Bmal1*-silenced HNSCC cell lines Cal-27 (**left**) and SCC-9 (**right**). (**A**,**B**) Histogram showing the *Bmal1* transcript level attained by q-RT-PCR in HNSCC cells transfected with siRNA-*Bmal1*. (**C**,**D**) The effect of melatonin on cell proliferation assessed by CyQuant assay. Data are presented as the mean ± SEM of *n* = 3 independent experiments; * *p* < 0.05, ** *p* < 0.01, *** *p* < 0.001 vs. mock control; # *p* < 0.05, ### *p* < 0.001 vs. siRNA-*Bmal1* control.

**Table 1 ijms-24-15030-t001:** Cosinor analysis of the relative expression of the clock genes *Bmal1* and *Per2* in HNSCC cell lines Cal-27 and SCC9 after serum shock, control (vehicle), and aMT treatments (500 and 1000 µM).

Cell Line	Gene	Treatment	Presence of Rhythmicity (*p*-Value)	Acrophase	Amplitude
Cal-27	*Bmal1*	Serum shock	<0.001	8.570589	0.492039
		Control	<0.01	10.76038	0.13079
		aMT 500 μM	<0.001	9.514851	0.225551
		aMT 1000 μM	Ns	-	-
	*Per2*	Serum shock	<0.01	23.05335	0.341575
		Control	Ns	-	-
		aMT 500 μM	Ns	-	-
		aMT 1000 μM	<0.05	4.777426	0.256121
SCC9	*Bmal1*	Serum shock	<0.001	5.105071	0.580572
		Control	<0.001	10.4789	0.211801
		aMT 500 μM	<0.01	10.58747	0.215423
		aMT 1000 μM	<0.001	12.57654	0.245783
	*Per2*	Serum shock	<0.001	22.4778	0.396895
		Control	Ns	-	-
		aMT 500 μM	Ns	-	-
		aMT 1000 μM	<0.001	9.562281	0.369996

**Table 2 ijms-24-15030-t002:** Circadian rhythm comparison of the relative expression of clock genes *Bmal1* and *Per2* in HNSCC cell lines Cal-27 and SCC9. * *p* < 0.05, ** *p* < 0.01, *** *p* < 0.001.

Cell Line	Gene	Comparison	*p*-Value for Acrophase Difference	*p*-Value for Amplitude Difference
Cal-27	*Bmal1*	Serum shock vs. Control	ns (0.328288)	** (0.002598)
		Serum shock vs. aMT 500 μM	ns (0.506968)	* (0.029825)
		Control vs. aMT 500 μM	ns (0.407893)	ns (0.182816)
	*Per2*	Serum shock vs. aMT 1000 μM	ns (0.012848)	ns (0.613012)
SCC9	*Bmal1*	Serum shock vs. Control	*** (2.95 × 10^−5^)	*** (0.000234)
		Serum shock vs. aMT 500 μM	*** (7.57 × 10^−5^)	*** (0.000461)
		Serum shock vs. aMT 1000 μM	*** (3.8 × 10^−8^)	*** (0.00063)
		Control vs. aMT 500 μM	ns (0.940019)	ns (0.968606)
		Control vs. aMT 1000 μM	ns (0.125217)	ns (0.693895)
		aMT 500 μM vs. aMT 1000 μM	ns (0.176089)	ns (0.738686)
	*Per2*	Serum shock vs. aMT 1000 μM	ns (1.25 × 10^−15^)	ns (0.653382)

**Table 3 ijms-24-15030-t003:** Cosinor analysis of the relative expression of *Sirt1* in HNSCC cell lines Cal-27 and SCC9 after serum shock, control, and aMT (500 and 1000 µM) treatments.

Cell Line	Gene	Treatment	Presence of Rhythmicity (*p*-Value)	Acrophase	Amplitude
Cal-27	*Sirt1*	Serum shock	<0.05	8.240747	0.18581
		Control	Ns	-	-
		aMT 500 μM	<0.01	10.27415	0.208889
		aMT 1000 μM	<0.01	12.19064	0.193523
SCC9	*Sirt1*	Serum shock	<0.01	7.629646	0.193267
		Control	Ns	-	-
		aMT 500 μM	<0.05	11.1633	0.154587
		aMT 1000 μM	<0.05	9.559829	0.174447

**Table 4 ijms-24-15030-t004:** Circadian rhythm comparison of the relative expression of *Sirt1* in HNSCC cell lines Cal-27 and SCC9.

Cell Line	Gene	Comparison	*p*-Value for Acrophase Difference	*p*-Value for Amplitude Difference
Cal-27	*Sirt1*	Serum shock vs. aMT 500 μM	ns (0.303475)	ns (0.832268)
		Serum shock vs. aMT 1000 μM	ns (0.073515)	ns (0.943527)
		aMT 500 μM vs. aMT 1000 μM	ns (0.271717)	ns (0.874045)
SCC9	*Sirt1*	Serum shock vs. aMT 500 μM	ns (0.087927)	ns (0.695747)
		Serum shock vs. aMT 1000 μM	ns (0.290038)	ns (0.842563)
		aMT 500 μM vs. aMT 1000 μM	ns (0.449046)	ns (0.844595)

**Table 5 ijms-24-15030-t005:** Cosinor analysis of OCR in HNSCC cell line Cal-27 after serum shock, control, and aMT (500 and 1000 µM) treatments.

Cell Line	Treatment	Presence of Rhythmicity (*p*-Value)	Acrophase	Amplitude
Cal-27	Serum shock	<0.001	17.00138	0.311756
	Control	Ns	-	-
	aMT 500 μM	Ns	-	-
	aMT 1000 μM	Ns	-	-
	Serum shock after control	<0.05	14.65534	0.143189
	Serum shock after aMT 1000 μM	<0.01	15.15964	0.225529

**Table 6 ijms-24-15030-t006:** Circadian rhythm comparison of OCR in HNSCC cell line Cal-27.

Cell Line	Comparison	*p*-Value for Acrophase Difference	*p*-Value for Amplitude Difference
Cal-27	Serum Shock vs. Serum shock after vehicle	ns (0.391999)	Ns (0.14612)
	Serum shock vs. Serum shock after 1000 μM aMT	ns (0.293366)	Ns (0.462536)

**Table 7 ijms-24-15030-t007:** Forward and reverse sequences of the primers used for PCR.

Gene	Gene Description	Forward Primer	Reverse Primer
*Bmal1*	Brain and muscle aryl hydrocarbon receptor nuclear translocator (ARNT)-like 1	ATCCTCAACTACAGCCAGAATG	TCGTGCTCCAGAACATAATCG
*Per2*	Period circadian clock 2	CCCTTCCGCATGACGCCCTACCTG	GACCGCCCTTTCATCCACATCCTG
*Sirt1*	Sirtuin 1	ACAGGTTGCGGGAATCCAAA	GTTCATCAGCTGGGCACCTA
*GAPDH*	Glyceraldehyde-3-phosphate dehydrogenase	GTACTACACTGAATTCACCCCCACTG	TGCGGCATCTTCAAACCTCCAT

## Data Availability

Data are available from the corresponding author upon request.

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
