# Peer review of "The Relationship between Clock Genes, Sirtuin 1, and Mitochondrial Activity in Head and Neck Squamous Cell Cancer: Effects of Melatonin Treatment"

_ijms, 2023, doi:10.3390/ijms241915030_

Round 1
Reviewer 1 Report
This article by César Rodríguez-Santana et al. is an interesting contribution to the study of the role of clock genes on mitochondrial dysfunction and cancer development. It is original since it studies, for the first time, the effect of high doses of melatonin on clock gene expression in HNSCC cell lines. It is well organized and the methodology is completely described. I consider this article is apt for publication. However, I suggest some minor changes which I list below.
1) Consider writing the 2 in CO2 (line 88) and O2 (line 120) as subscript: CO2 and O2.
2) Consider unifying the way of expressing the concentrations of melatonin in the abstract (0.5 and 1 mM) with that of the rest of the manuscript (500 and 1000 µM). Even when they are equivalent, expressing them identically across the manuscript can ease understanding.
3) Consider unifying the way of mentioning genes. There are some instances in which Bmal1 and Per2 appear with initial capital letters (Eg Table 2, lines 168, 179, 182, 191, 197, etc) while in some others they do not (line 146). The same happens with Sirt1 (line 211) vs. sirt1 in table 5.
Author Response
Response to reviewer´s comments
First, we would like to return the revised manuscript entitled” “Relationship between clock genes, Sirtuin 1 and mitochondrial activity in head and neck squamous cell cancer: effects of melatonin treatment” (Manuscript ID ijms-2634362) to International Journal of Molecular Sciences for its consideration of publication. We would like to thank reviewers for critically examining this manuscript and providing us invaluable comments and suggestions. We respond to each of the reviewer’s comments and clarified/amended every necessary part. The statements addressing the referee's comments point by point are listed below and the revised text with alterations in red font was uploaded. We highly appreciate your consideration and further process.
Amendments to the questions raised by the reviewer #1 to the manuscript ID ijms-2634362
This article by César Rodríguez-Santana et al. is an interesting contribution to the study of the role of clock genes on mitochondrial dysfunction and cancer development. It is original since it studies, for the first time, the effect of high doses of melatonin on clock gene expression in HNSCC cell lines. It is well organized and the methodology is completely described. I consider this article is apt for publication. However, I suggest some minor changes which I list below.
1) Consider writing the 2 in CO2 (line 88) and O2 (line 120) as subscript: CO2 and O2.
-Corrected
2) Consider unifying the way of expressing the concentrations of melatonin in the abstract (0.5 and 1 mM) with that of the rest of the manuscript (500 and 1000 µM). Even when they are equivalent, expressing them identically across the manuscript can ease understanding.
-Corrected
3) Consider unifying the way of mentioning genes. There are some instances in which Bmal1 and Per2 appear with initial capital letters (Eg Table 2, lines 168, 179, 182, 191, 197, etc) while in some others they do not (line 146). The same happens with Sirt1 (line 211) vs. sirt1 in table 5.
-Corrected

Reviewer 2 Report
Why did you choose these cell lines? Please describe the nature of the cell lines and state why you chose these over others.
Can you add more cell lines to support your findings?
Why were these dosages of melantonin chosen? - Can you show a titration curve?
What is the rationale for stating that there is an antiproliferative effect in the cell lines?
"our results suggest that clock genes are not involved in the cytotoxic activity of melatonin in HNSCC" - in these cell lines.
Good quality.
Author Response
Response to reviewer´s comments
First, we would like to return the revised manuscript entitled” “Relationship between clock genes, Sirtuin 1 and mitochondrial activity in head and neck squamous cell cancer: effects of melatonin treatment” (Manuscript ID ijms-2634362) to International Journal of Molecular Sciences for its consideration of publication. We would like to thank reviewers for critically examining this manuscript and providing us invaluable comments and suggestions. We respond to each of the reviewer’s comments and clarified/amended every necessary part. The statements addressing the referee's comments point by point are listed below and the revised text with alterations in red font was uploaded. We highly appreciate your consideration and further process.
Amendments to the questions raised by the reviewer #2 to the manuscript ID ijms-2634362
1) Why did you choose these cell lines? Please describe the nature of the cell lines and state why you chose these over others.
Both the Cal-27 and SCC9 cell lines represent a model widely used in the study of human head and neck cancer, whose tissue of origin corresponds to the tongue of 56- and 25-year-old males, respectively. They have even been used for in vitro and in vivo studies to study circadian machinery previously (Chen et al., doi.org/10.1038/srep24324; Tang et al., doi.org/10.1158/0008-5472.CAN-16-1322). In this context, we have previously demonstrated that melatonin reverses metabolic reprogramming and decreases proliferation and increases apoptosis in HNSCC cell lines CAL-27 and SCC9. Moreover, we have demonstrated in these two cell lines that melatonin enhances the cytotoxic effects of rapamycin, cisplatin and radiotherapy in HNSCC cells CAL-27 and SCC9 (Shen YQ et al, doi.org/10.1111/jpi.12461; Fernandez-Gil BI et al, doi.org/10.1155/2019/7187128; Guerra-Librero A et al, doi.org/10.3390/antiox10040603; Florido et al, doi.org/10.1111/jpi.12824)
A summary of this information was added to the introduction section (page 2):
“In In this context, we have previously demonstrated that melatonin reverses metabolic reprogramming and decreases proliferation in HNSCC cell lines CAL-27 and SCC9”
2) Can you add more cell lines to support your findings?
Unfortunately, we cannot provide results in other cell lines since the study of the expression of clock genes is an area that requires a large number of experiments and time to obtain conclusive data.
3) Why were these dosages of melatonin chosen? - Can you show a titration curve?
Previously, we performed a dose response study, and our data suggested that there is a correlation between the cell content of melatonin and its apoptotic effects, thus supporting the notion that high concentrations of melatonin in cancer cells are required to its cytotoxic effects (14, 15). However, this study aims to clarify whether the effects observed and published by our research group, effects mainly targeting mitochondria, are due to a regulation of this hormone in the circadian machinery. For this reason, we considered the use of the same doses of melatonin used previously and even, to clarify an effect of melatonin at lower concentrations, we decided to include the groups treated at 10 nM and 100 µM.
A summary of this information was added to the Results section (page 4).
4) What is the rationale for stating that there is an antiproliferative effect in the cell lines?
In previous study, we showed that melatonin decreases proliferation in a dose dependent-manner in the Cal-27 and SCC9 cell lines and enhances the cytotoxic effects of rapamycin, CDDP and radiotherapy. In this article we affirm that the antiproliferative effect of melatonin is not dependent on Bmal1. To do this, we used the CyQUANT assay and compared the effect of melatonin at doses of 500 and 1000 uM in Mock cells as well as in those siRNA-Bmal1 cells. The data reflected a dose-response decrease in cell proliferation in both cell groups for both cell lines. Furthermore, an increase in proliferation is observed in the siRNA-Bmal1 control group compared to the Mock control group. This is probably explained by the close regulation of Bmal1 in the regulation of the cell cycle (Geyfman et al., doi.org/10.1073/pnas.1209592109; Matsumoto et al., doi.org/10.18632/oncotarget.9877; Tang et al., doi.org/10.1158/0008-5472.CAN-16-1322), so a decrease in this protein could trigger an increase in proliferation. Therefore, the decrease in proliferation in a dose-dependent manner at melatonin concentrations indicates that the antiprolifrative effect of melatonin in the Cal-27 and SCC9 cell lines is not due to an effect of this hormone on Bmal1.
5) "our results suggest that clock genes are not involved in the cytotoxic activity of melatonin in HNSCC" - in these cell lines.
-Corrected in page 10

Round 2
Reviewer 2 Report
Thank you for revising.